# Revealing the Metabolic Alterations during Biofilm Development of *Burkholderia cenocepacia* Based on Genome-Scale Metabolic Modeling

**DOI:** 10.3390/metabo11040221

**Published:** 2021-04-05

**Authors:** Ozlem Altay, Cheng Zhang, Hasan Turkez, Jens Nielsen, Mathias Uhlén, Adil Mardinoglu

**Affiliations:** 1Science for Life Laboratory, KTH–Royal Institute of Technology, 171 65 Solna, Sweden; oaltay@kth.se (O.A.); chengzh@kth.se (C.Z.); mathias.uhlen@scilifelab.se (M.U.); 2Department of Clinical Microbiology, Sami Ulus Training and Research Hospital, University of Health Sciences, Ankara 06080, Turkey; 3School of Pharmaceutical Sciences, Zhengzhou University, Zhengzhou 450001, China; 4Department of Medical Biology, Faculty of Medicine, Atatürk University, Erzurum 25 240, Turkey; hturkez@atauni.edu.tr; 5Department of Biology and Biological Engineering, Chalmers University of Technology, 412 96 Gothenburg, Sweden; nielsenj@chalmers.se; 6Centre for Host-Microbiome Interactions, Faculty of Dentistry, Oral & Craniofacial Sciences, King’s College London, London SE1 9RT, UK

**Keywords:** *Burkholderia cenocepacia*, biofilm, genome-scale metabolic models, synthetic lethality, transcriptomics, omics integration

## Abstract

*Burkholderia cenocepacia* is among the important pathogens isolated from cystic fibrosis (CF) patients. It has attracted considerable attention because of its capacity to evade host immune defenses during chronic infection. Advances in systems biology methodologies have led to the emergence of methods that integrate experimental transcriptomics data and genome-scale metabolic models (GEMs). Here, we integrated transcriptomics data of bacterial cells grown on exponential and biofilm conditions into a manually curated GEM of *B. cenocepacia*. We observed substantial differences in pathway response to different growth conditions and alternative pathway susceptibility to extracellular nutrient availability. For instance, we found that blockage of the reactions was vital through the lipid biosynthesis pathways in the exponential phase and the absence of microenvironmental lysine and tryptophan are essential for survival. During biofilm development, bacteria mostly had conserved lipid metabolism but altered pathway activities associated with several amino acids and pentose phosphate pathways. Furthermore, conversion of serine to pyruvate and 2,5-dioxopentanoate synthesis are also identified as potential targets for metabolic remodeling during biofilm development. Altogether, our integrative systems biology analysis revealed the interactions between the bacteria and its microenvironment and enabled the discovery of antimicrobial targets for biofilm-related diseases.

## 1. Introduction

Cystic fibrosis (CF) is the most common form of the autosomal recessive disease in Caucasians and affects 1 in every 3000–3500 births in Europe and United States [1]. CF Foundation’s National Patient Registry reports the median age of survival is 33.4 years and respiratory diseases are the leading cause of early mortality in people with CF [2]. Many risk factors are implicated in the development and progression of CF lung disease, such as chronic infections by biofilm-forming pathogens. Complex metabolic responses in biofilms may have a fundamental role in the pathogenesis of this chronic disease [3].

*Burkholderia cepacia* complex (Bcc) emerged as important pathogens in the airways of immunocompromised humans, especially those with CF. The treatment of Bcc infections is challenging because of the natural resistance to various classes of antibiotics and long-term colonization, and the ability to adapt to nutrient-deficient biofilm microenvironment [4]. There is an urgent need to develop efficient drugs to improve and preserve lung function early on in patients’ life since chronic infection with Bcc is an independent predictor associated with poor prognosis [5]. The eradication of Bcc remains challenging, and there are large gaps in our current understanding of the underlying biochemical and molecular mechanisms involved in the persistence of biofilms.

Revealing the pathogen and microenvironment interactions requires an analysis of large reaction networks, which can be identified through the use of systems biology approaches. Systems biology is an integrative discipline that assembles experimental data with computational methods to describe the sophisticated biological processes of cells, tissues, and microorganisms [6]. Genome-scale metabolic models (GEMs) are denominators of systems biology and are defined as the computational reconstruction of entire biochemical reactions known to occur in an organism [7]. These models have been used to integrate experimentally derived knowledge to reveal the underlying molecular mechanisms of interrelated metabolic processes associated with a condition (e.g., the biofilm formation that directs antimicrobial resistance and chronic infections) [8,9,10].

In this study, we investigated the metabolic interactions between *B. cenocepacia* with its microenvironment based on the metabolic modeling and flux balance analysis (FBA) approach. Next, we examined the influence of uptake reactions, which involve substrate exchange with the extracellular space, not only on bacterial growth rates but also on flux distribution across intracellular pathways based on synthetic lethality analysis. Afterward, gene expression data from two different conditions (exponential and biofilm) were integrated with the GEM of *B. cenocepacia* to investigate the effect of environmental conditions on bacterial metabolism.

## 2. Methods

### 2.1. Genome-Scale Metabolic Model of Burkholderia Cenocepacia

The GEM of *Burkholderia cenocepacia* J2315 (iPY1537) was downloaded in SBML format from the authors’ website (https://bme.virginia.edu/csbl/Downloads1-Burkholderia-cepacia.html, Access Date: 29 October 2019) [11]. Details regarding the metabolic model of the microorganism are provided in Appendix A. We performed model refinement by manual curation to enable more consistent data with biological functionality. The COBRA Toolbox (Constraint-Based Reconstruction and Analysis Toolbox) was used to design, analyze, and predict the perturbations on the models [12]. Biomass production was defined as the objective function to predict the proportion of metabolic pathway usage through the FBA. The COBRA Toolbox was implemented in MATLAB 2019a (The MathWorks, Inc., Natick, MA, USA), and academic licenses of IBM CPLEX v12.10.1 (IBM, Armonk, NY, USA) were used to solve linear optimization problems in this study.

### 2.2. Synthetic Lethality Analysis

We performed the simulation of single and double synthetic lethal reactions by constraining upper and lower bounds of the fluxes to zero and selecting the constraint that would make the objective function unfeasible. Individual reactions that inhibit the objective function considered as single lethal reactions. Constraining the model by removing one pair of reactions at a time (excluding single lethal reactions), which resulted in no cell growth, was determined as double lethal reactions. Exchange reactions were settled by available COBRA functions and were constrained individually, and were paired by setting these fluxes to zero. Biomass formation was used as the objective function. Essential exchange reactions were defined as those in which the absence of nutrients by inhibition of these reactions eventually leads to cell death. Cellular metabolic pathways were determined from enclosed subsystems of the model, and unassigned pathways were disregarded.

### 2.3. Differential Expression Analysis of RNA-Seq Data

The raw RNA sequencing data of wild type *B. cenocepacia* J2315 isolates under exponential and biofilm conditions were downloaded from ArrayExpress (accession number: E-MTAB-5526, ENA Project Number: PRJEB19669) [13]. Six RNA-seq datasets include three biological replicates of exponential and biofilm grown *B. cenocepacia*. Sample accession numbers of the transcripts are accessible in Appendix A.

We quantified gene expression levels with the alignment-free pipeline Kallisto (v0.46.1) [14], which determines the compatibility of high-throughput sequencing reads with target sequences by using k-mer-based counting algorithms. We used RNA-seq fastq read data and transcriptome reference fasta data of *B. cenocepacia* J2315 [15]. Initially, binary index from cDNA transcripts in FASTA format was built by using *index* function. Then, estimation of the transcripts’ abundances was performed with *quant* function. The abundance estimates were reported in transcripts per million (TPM).

Gene level differential expression data were analyzed with the DESeq2 package (v1.22.2) [16] in the R platform (v3.5.3). This method estimates the variance–mean dependence in count data from high-throughput sequencing assays and test for differential expression based on a model using the negative binomial distribution. Raw counts of sequencing reads in the form of a matrix of integer values were obtained with a phenodata file describing the experimental groups. The output of this analysis was the significantly differentially expressed genes. 

### 2.4. Integrating RNA-Seq Data with the Genome-Scale Metabolic Model

We integrated transcriptomics data with the GEM of *B. cenocepacia* by the Metabolic Adjustment for Differential Expression (MADE) algorithm using Toolbox for Integrating Genome-Scale Metabolic Models, Expression Data, and Transcriptional Regulatory Networks (TIGER) package [17,18]. This is a consistent platform for algorithm development and extending existing GEMs with regulatory networks and high-throughput data. This algorithm accounts for the expression state of a gene with a weighted consideration of the statistical significance of the gene expression changes. TIGER converts a gene–protein relation (GPR) and additional regulatory rules into an equivalent mixed integer linear program (MILP). The MILP constraints are added to a COBRA metabolic model to create a TIGER model that combines metabolism, GPR associations, and transcriptional regulation. Condition-specific models were generated for the exponential and biofilm phase of *B. cenocepacia*. To understand how the transition from exponential phase to biofilm phase impacts the intracellular metabolic fluxes and growth of *B. cenocepacia*, we performed the flux variability analysis of all fluxes reaching the optimal solution. Constraints were defined according to synthetic cystic fibrosis sputum medium (SCFM), a medium designed to replicate the environment of the cystic fibrosis lung [19]. Minimum and maximum fluxes were calculated for all reactions through the built-in flux variability analysis (FVA) function of the TIGER package. The defined subsystems on the model iPY1537 were used to identify the most deregulated pathways within the transition from exponential to biofilm conditions (Figure 1).

## 3. Results

### 3.1. In Silico Identification of Reaction Essentialities and Affected Pathways

To assess the in silico identification of different growth requirements, we retrieved GEM of *B. cenocepacia* (iPY1537), which contained 1667 reactions and 1513 genes. We evaluated the influence of a reaction without flux on the entire known metabolic network through the synthetic lethality analysis approach (Figure 2A). This allowed for identifying 174 single essential reactions and 115 combinations of reaction pairs excluding single lethal reactions (Figure 2B). Reactions involved in lipid metabolism constitute the majority (*n* = 103, 59.2%) of the single lethal reactions; among these lipid metabolism reactions, fatty acid biosynthesis (*n* = 62, 60.2%) was most frequently found. In remaining 71 single lethal reactions, glycan metabolism reactions were the most common (*n* = 37, 52.1%) (Figure 2B, Appendix A). 

Response to microenvironment nutrient status changes was evaluated by blocking exchange or transport reactions (*n* = 162) of the model. Notably, as the exchange and transport reactions in the model of *Burkholderia cenocepacia* J2315 have been extensively curated based on medium information, we grouped them together in our essentiality analysis to investigate the potentially important metabolites in the environment for the growth of bacteria. Exchange of oxygen and O_2_ transport via diffusion were identified as single lethal exchange/transport reactions (Figure 2C, Appendix A).

Considering pathways in all lethal pairs, reactions belonging to the pyrimidine metabolism (*n* = 43) were most commonly responsible for pairs becoming lethal and were found in 26.1% of the pairs (Figure 2B, Appendix A). ATP nucleoside-diphosphate phosphotransferases were dominant enzymes of pyrimidine metabolism related lethal reaction pairs. In all model, different types of phosphotransferases were involved in 44% of all lethal reaction pairs (Appendix A).

The model of *B. cenocepacia* presented 67 pairs of lethal combinations with at least one exchange or transport reaction (Figure 2C). Reactions in phenylalanine, tyrosine, and tryptophan biosynthesis (*n* = 22, 32.8%) were most common in lethal pairs with at least one exchange/transport reaction, and paired with tryptophan supply of the cell (Figure 2B, Appendix A). Tryptophan is a member of aromatic amino acids and has been linked to various metabolic functions involved in the maintenance of redox homeostasis and NAD+ biosynthesis [20]. Valine, isoleucine, cytidine, and deoxyguanosine were the other frequent compounds that absence with an additional reaction inhibits the entire metabolic network (Figure 2C, Appendix A).

### 3.2. Condition-Dependent Metabolic Models of B. cenocepacia

We analyzed the physiology of *B. cenocepacia* during adaptation to the biofilm environment based on transcriptomics data [13] of cells collected in exponential phase and biofilm phase. To integrate the comparative gene-expression data from exponential versus biofilm conditions into iPY1537, we produced contextualized metabolic models for the exponential phase and the biofilm environment. The 602 (72.1%) out of a total of 835 differentially expressed genes could be consistently included in MADE. Our analysis indicated that 228 genes were upregulated, 216 genes were downregulated, and 158 genes were constant in the transition from exponential to biofilm phase. Objective fluxes were calculated as 11.66 and 11.60 mmol/gDW/h for exponential and biofilm conditions after expression data was applied.

Flux variability analysis results revealed that 590 reactions had flux at least one condition. Among those, 303 reactions (51.4%) were in the same directionality, and 287 reactions (48.6%) were altered between conditions. We found that a total of 176 unique exchange/transport reactions, 79 (44.9%) reactions were common, and 97 (55.1%) reactions varied across two different conditions (Appendix A). Reactions involved in glycerolipid synthesis (100%), lipopolysaccharide biosynthesis (100%), fatty acid biosynthesis (98.4%), and glycerophospholipid metabolism (90.4%) were the most conserved reactions in both conditions (Figure 3, Appendix A). 

Metabolism of amino acids with 59 (31%) altered reactions scored as the most varied pathway when we excluded exchange/transport reactions. This was followed by carbohydrate metabolism reactions with 52 (27%) and purine and pyrimidine metabolism with 29 (15%) reactions that introduced changes. Vitamin metabolism was the fourth most altered pathway with 20 (11%) changing reactions in total for ascorbate and aldarate metabolism, folate metabolism, and pantothenate and CoA biosynthesis (Appendix A).

In detail, pathways in arginine and proline metabolism (100%) and glycine, serine, and threonine metabolism (100%) and ascorbate and aldarate metabolism (100%) were the pathways, in which all the reactions are altered between two conditions (Figure 4). This is followed by alanine, aspartate, and glutamate metabolism (91%) and valine, leucine, and isoleucine metabolism (77%), in which most of the reactions are changed in biofilm conditions (Figure 4). The model predicted that direct conversion of serine to pyruvate and conversion of glutamate to 2-oxaloacetate through producing phosphoserine is only active in biofilm conditions. 

We also examined the regulation of vitamin metabolism and the subsequent effect on various physiological functions in the system during biofilm conditions. We found that both arginine proline metabolism and the ascorbate aldarate pathway are connected to the citrate cycle through 2,5-dioxopentanoate synthesis.

Condition-specific models had fluxes on 119 and 157 exchange/transport reactions for exponential and biofilm conditions, respectively. In both phases, a total of 97 fluxes were unchanged. According to the altered exchange/transport reactions, more amino acids were consumed when the cells were at the biofilm phase: 3 and 14 additional amino acid intakes were predicted in addition to common exchange/transport reactions in exponential and biofilm conditions, respectively (Appendix A). As the carbohydrate reserve was depleted, amino acid breakdown outweighs, and, eventually, the accessible nutrients were obtained from the dead bacteria. To sustain growth in these circumstances, bacteria should adjust to an elevated energy cost for the catabolism of the biomolecules from the bacterial debris.

## 4. Discussion

In this study, the systems biology approach for model perturbation and data integration allowed us to evaluate how environmental conditions affect the metabolic pathways of *B. cenocepacia* J2315. Altered metabolic fluxes were identified from the synthetic lethality analysis as well as condition-specific models based on gene expression profile. The models predicted tendencies to exclusive pathways in the biofilm development when comparing exponential conditions.

First, we provided a comprehensive analysis of essential reactions and affected pathways in *B. cenocepacia* under exponential conditions. The influence of different conditions on synthetic lethal interactions was adjusted by perturbing the model based on reaction constraints. According to the synthetic lethality analysis, reactions involved in fatty acid biosynthesis were most frequently found in single lethal reactions. Control at the level of fatty acid biosynthesis is crucial for membrane homeostasis, because the biophysical properties of membranes are determined by the composition of the fatty acids that are produced by de novo biosynthesis. Since lipids are the cell’s primary structural components, the enzymes of bacterial fatty acid biosynthesis are attractive targets for antimicrobial drug discovery [21]. To this extent, various inhibitors of bacterial fatty acid biosynthesis were developed; however, only two of those (e.g., isoniazid and triclosan) are in clinical usage [22]. We claim that a combination of rational drug design methodologies with GEM predictions of distinct metabolic pathways will have more success on selectively targeted drug development.

Reactions involved in phenylalanine, tyrosine, and tryptophan biosynthesis were most frequently found in lethal pairs with at least one exchange/transport reaction when we excluded single lethal reactions. Tryptophan exchange was responsible for 36% of those reaction pairs becoming lethal. In this manner, Zhang et al. showed that the absence of exogenous tryptophan caused an immediate killing on auxotrophic mycobacterium species, which is consistent with our findings [23]. Since tryptophan is not synthesized by human cells, biosynthetic pathways of this amino acid provide excellent targets for the discovery of new antimicrobial agents [24].

Second, to obtain a holistic understanding of the interactions between *B. cenocepacia* with its environment and how the metabolic network is regulated as a result of biofilm development, we integrated the transcriptomics data of exponential and biofilm phases into the iPY1537. Our results highlighted that glycerolipid synthesis, lipopolysaccharide biosynthesis, fatty acid biosynthesis, and glycerophospholipid metabolism were least affected when the conditions are altered. Bacterial lipid metabolism is responsible for producing structural molecules and biological energy storage, and the biosynthesis of the extracellular polymeric substances that are crucial for biofilm formation [25]. Phospholipid biosynthesis is an essential component for the formation and maintenance of the membrane, which is initiated with the synthesis of the fatty acids. Although fatty acid synthesis is a complex and energy-intensive process, there are no alternate aspects to fulfill their central roles in membrane formation. Therefore, the fundamental process of lipid metabolism is highly conserved in bacteria among different life conditions [25]. The encouraging results obtained in this study facilitate the understanding of the mechanisms that regulates the pathway usage across conditions. 

While earlier studies have prospected the organization of amino acid and carbohydrate metabolism of microorganisms under different environmental conditions [26,27], the GEM reconstruction of *B. cenocepacia* allowed us to investigate these features in the context of the metabolic modeling. Integration of transcriptomic data of the biofilm conditions revealed that bacteria tend to uptake more amino acids from the environment for growth, and reactions of the pentose-phosphate pathway are activated. These in silico predictions are compatible with the fact that throughout exponential growth, carbohydrates are first consumed as the main supply of carbon and energy but are promptly exhausted, and additional fuel molecules such as lipids and proteins are utilized to maintain survival [28].

Pathway-based analysis also unveiled which fluxes in amino acid metabolism were most affected by biofilm conditions. Glycine, serine, and threonine metabolism and arginine and proline metabolism were the two subsystems in which all of the reactions were altered between conditions. In particular, the glycine, serine, and threonine pathway is closely related to central metabolism. The amino acid serine is an indispensable biological molecule, either a basic constituent for protein synthesis or as a precursor of other amino acids, nucleotides, and phospholipids [29]. In parallel, Greenwich et al. revealed that serine levels declined in the stationary phase compared to the exponential phase, and expression of genes involved in the serine biosynthesis downregulated when *Bacillus subtilis* entered the stationary phase [30]. They also showed that the deletion of the serine deaminase gene resulted in a deceleration in biofilm organization, which endorsed the idea that serine levels are essential for biofilm formation. Other previous studies have also indicated that serine is consumed more rapidly than other amino acids in *Escherichia coli* and almost complete depletion of serine in the stationary phase [31,32,33,34]. Despite the biochemical processes that occur within serine metabolism are widely established, many aspects of the regulation of the serine homeostasis and major mechanisms to maintain the intracellular serine concentrations in bacteria remain incompletely defined. Our model calculations (e.g., direct conversion of serine to pyruvate and conversion of glutamate to 2-oxaloacetate through producing phosphoserine are only active in biofilm conditions but not in the exponential phase) agree with these literature findings and provide mechanistic explanations for the association between serine homeostasis and biofilm development in *B. cenocepacia*. It can be assumed that serine is first converted to pyruvate, then used for gluconeogenesis or channeled into the Krebs cycle for energy production. Therefore, antimetabolite for serine can be potentially used to block these serine related metabolic pathways and inhibit the growth or biofilm formation, which should be evaluated in future studies.

Several reactions in ascorbate and aldarate metabolism, folate metabolism, and pantothenate biosynthesis were altered, based on our metabolic analysis. These reactions are involved in various biological processes as the energy supply for growth and survival. In this manner, the role of the 2,5-dioxopentanoate and related pathways offer targets for understanding the aspects of microbial biofilm metabolism.

In this study, we performed computational analysis to reveal the interactions between the global properties of *B. cenocepacia* with its microenvironment and predicted the physiology during adaptation to the biofilm condition, which allowed us prior knowledge to provide testable hypotheses regarding the functionality of metabolic pathways. We pointed out considerable properties in metabolic response to extracellular nutrient availability and pathway susceptibility to altered growth conditions. These predictions can guide further experimental validation in the context of characterizing essential linkages of the bacteria with their microenvironment and discovery of potential targets for biofilm-based infections. As the metabolic models only focused on metabolic genes (i. e. regardless of genes involved in regulatory processes), future studies may integrate the metabolic models together with other biological systems, such as regulatory, signaling, and protein–protein interaction networks to enable more comprehensive understanding of the biofilm formation process and pathogenic role of the bacteria.

## Figures and Tables

**Figure 1 metabolites-11-00221-f001:**
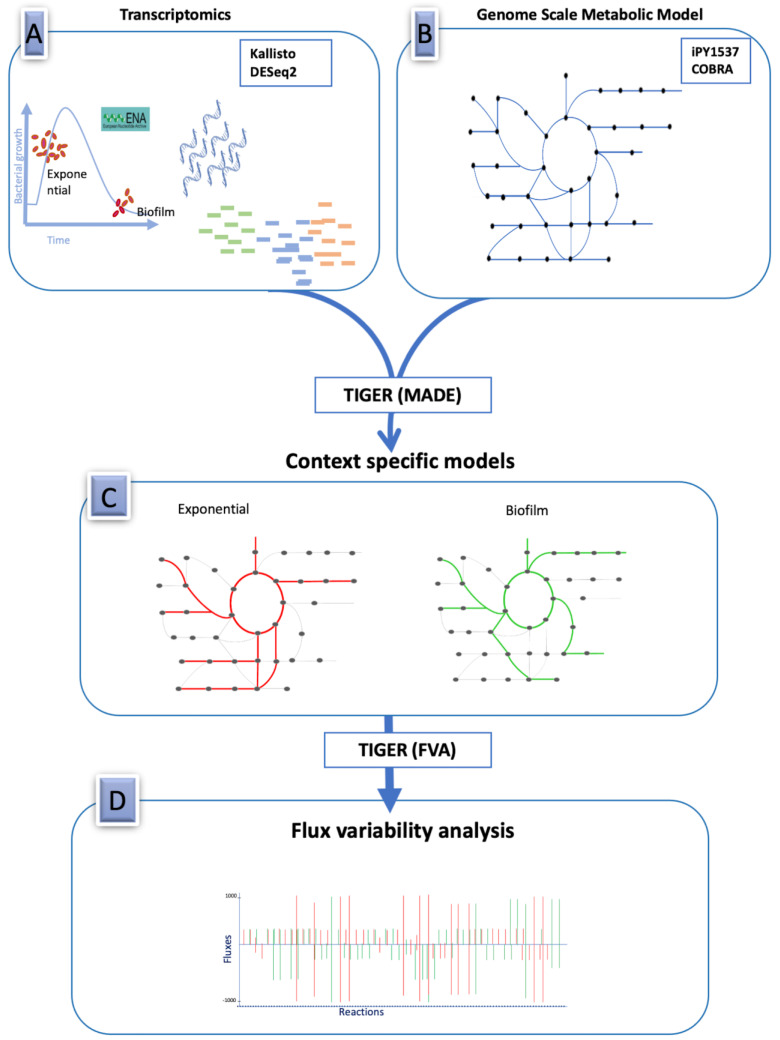
Study design for integration of transcriptomics data of *Burkholderia cenocepacia* into a manually curated genome-scale metabolic network. (**A**) Differential expression analysis of RNA sequencing data of bacterial isolates grown on exponential and biofilm conditions was performed using Kallisto and DESeq2. (**B**) Schematic representation of COBRA model for *Burkholderia cenocepacia* (iPY1537). (**C**) High-throughput gene expression data was integrated with genome scale model by the Metabolic Adjustment for Differential Expression (MADE) algorithm using Toolbox for Integrating Genome-Scale Metabolic Models, Expression Data, and Transcriptional Regulatory Networks (TIGER) package. (**D**) Schematic representation of minimum and maximum fluxes which were calculated for all reactions through the in-built flux variability analysis (FVA) function of the TIGER package. Red lines represent fluxes in exponential phase and green lines biofilm environment.

**Figure 2 metabolites-11-00221-f002:**
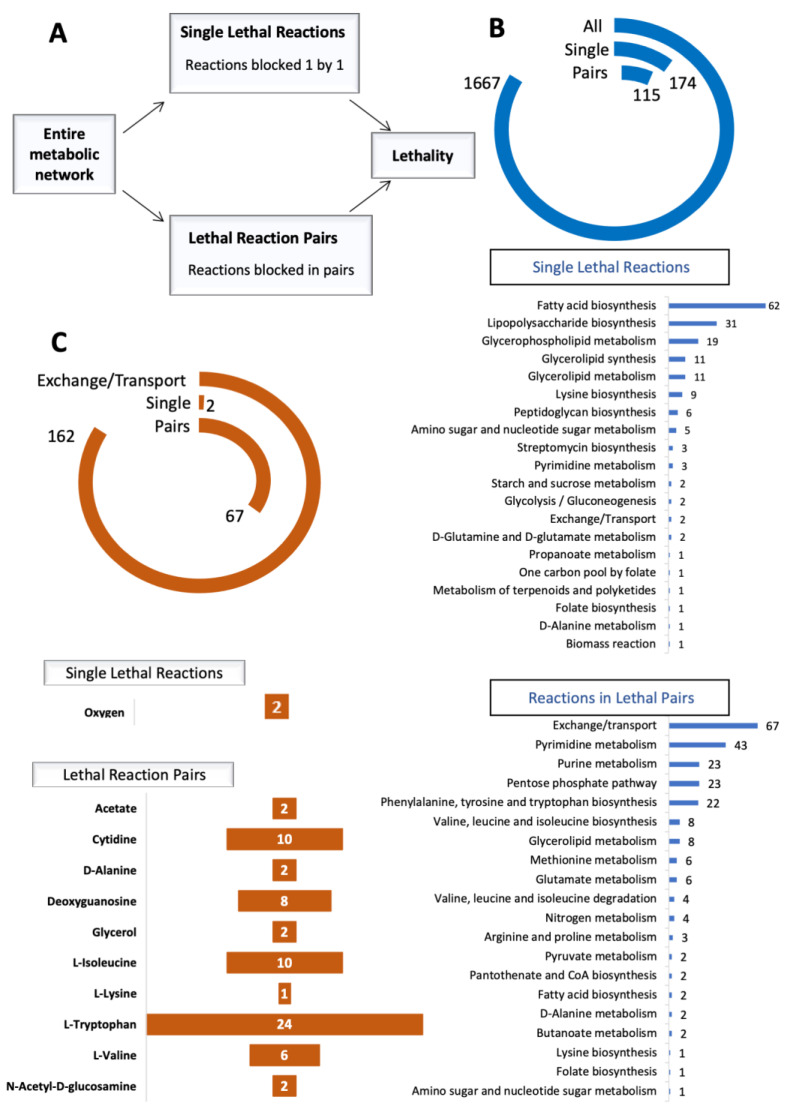
(**A**) Study design for synthetic lethality analysis. (**B**) Single lethal reactions, lethal reaction pairs, and related pathways. Blue circle plot shows numbers of single lethal reactions (Single), lethal reaction pairs (Pairs) and all reactions in the model (All). Chart bar of single lethal reactions shows numbers of involved reactions in each pathway. Chart bar of reactions in lethal pairs shows numbers of individual reactions in each pathway. (**C**) Single lethal exchange/transport reactions, exchange/transport reactions in lethal reaction pairs and related metabolites. Brown circle plot shows numbers of single lethal exchange/transport reactions (Single), exchange/transport reactions in lethal reaction pairs (Pairs), and all exchange/transport reactions in the model (Exchange/Transport). Bar chart of single lethal reactions shows numbers of exchange/transport reactions and related metabolites. Bar chart of lethal reaction pairs shows numbers of individual exchange/transport reactions and related metabolites.

**Figure 3 metabolites-11-00221-f003:**
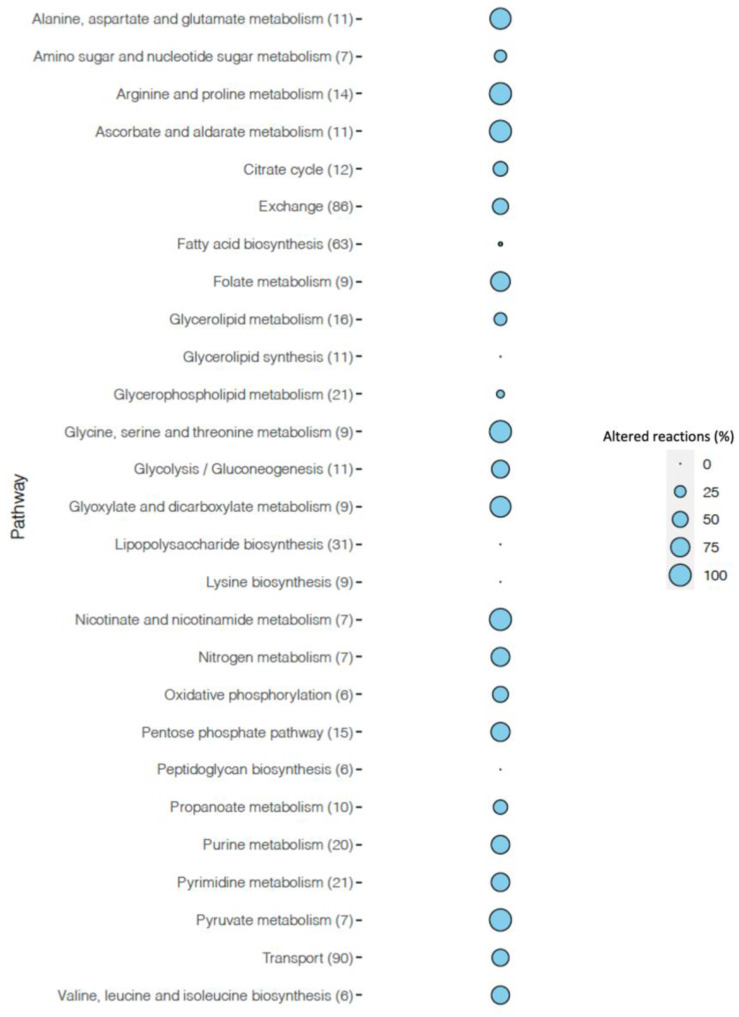
Balloon plot represents the percentage of altered reactions in pathways when transition from exponential phase to biofilm phase.

**Figure 4 metabolites-11-00221-f004:**
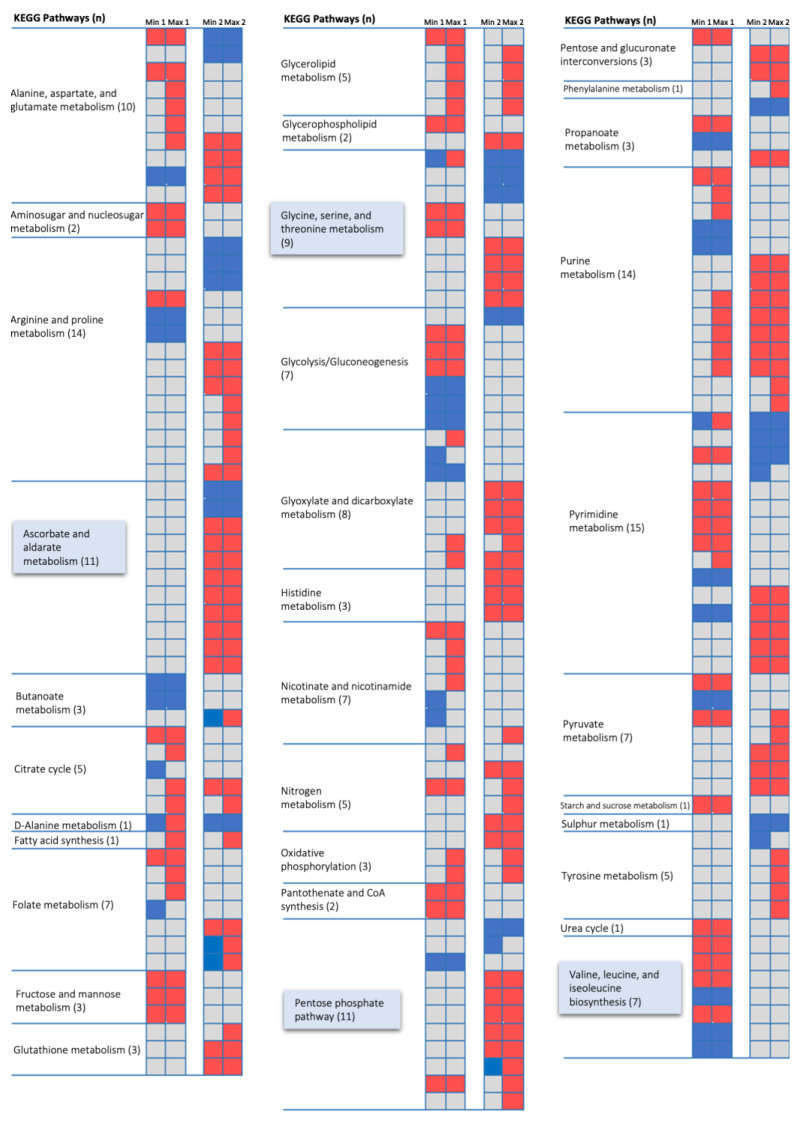
Detailed schematization of all altered reactions by using results of FVA. Red and blue colors represent the fluxes with positive and negative values, respectively. Gray represents no flux. Min 1, minimum flux in exponential condition; Max 1, maximum flux in exponential condition; Min 2, minimum flux in biofilm condition; Max 2, maximum flux in exponential biofilm. Highly altered pathways are highlighted in blue boxes; *n* is the number of reactions of the pathway that had flux for at least one condition.

## Data Availability

All code used for the analyses is available in https://github.com/hoaltay/BurkBiofilm (accessed on 4 April 2021).

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
