# Peer review of "Revealing the Metabolic Alterations during Biofilm Development of Burkholderia cenocepacia Based on Genome-Scale Metabolic Modeling"

_metabolites, 2021, doi:10.3390/metabo11040221_

Round 1

Reviewer 1 Report

I like this paper.  I think it is addressing an important problem in therapeutic development for CF patients.  I believe the basic of approach of the paper to be sound and I support the use of COBRA and TIGER etc.

I have several comments aimed at improving the paper.

First the figures, including the labels in the figures could be improved

Fig 1 shows the flow to get to the two conditions, but does not include the flow of the analysis after the TIGER runs on the two conditions.  It would be improved by including how the differential expression results, can be used to prioritize experiments and drug targets.

Fig 2 should spell out the headings for the single and double lethality results and in general make it easier to understand the results.

Fig 3 is also can be improved as the heat map sorting appears to be neither based on different or subsystem.  I encourage the authors to make the figure easier to understand and connect to the text.

In the paragraph that starts on line 203 contains important comparisons to the literature, it was how ever somewhat difficult to understand if there is a hypothesis being suggested that could be tested in B. cenocepacia.  I believe  a more direct statement of a testable hypothesis should be made here in comparison to the E. coli and B. subtilis results that at cited.  You relate these to the model but need to be more clear precisely which comparisons you think should be tested.

Fig 4 also has a problem with abbreviations in the columns and many rows that are associated with the subsystem not labeled.  One improvement would be to flag those differences that are interesting or counter intuitive that need further investigations.   Please consider how to improve this figure.

Section 4. I found the text in the discussion section to be disappointing.  I was looking for more concrete results that should be followed up by experimental work (hypotheses to confirm etc.).  Also the text English quality is not as good at the abstract and I think needs to be improved.

For work like this to have maximum impact, experimentalists should be able to digest the results and construct specific experiments to confirm or deny the hypotheses suggested by the computational analysis.  In this regard the paper is incomplete and the discussion section needs to be improved to make a set of prioritized predictions that need to be tested for the field to move forward.

GEMs always are partial reconstructions.  I would like to also see some discussion of genes and functions that are not included in the model that might have some bearing on the CF context and how those functions might be addressed in the future.

Reviewer 2 Report

To study the metabolic alterations duting biofilm development, Altay et al. integrated transcriptomics data (RNA-seq) of bacterial cells grown on exponential and biofilm conditions into manually curated GEM of B. cenocepacia J2315 (iPY1537).

The authors observed that during biofilm development, bacteria had altered pathway activities associated with several aminoacids and pentose phosphate pathways.

Major Comments:

1) Abstract, first phrase: Please rephrase this sentence. Burkholderia cepacia complex bacteria causes lower than 5% of the infections in CF patients. So is not the most prevalent.

2) Page 2, line 45, 46, 50 and 51: Please change Burkholderia spp to Burkholderia cepacia complex (Bcc) in the first line and Bcc in the other lines. Not all species of the genera Burkholderia causes infection in CF patients, but all Bcc can cause infection.

3) Figure 2B: In the circular representation are described 115 Jdl, however in the bottom representation are represented 230 Jdl. This information is confusing.

4) Page 5, line 154: Please confirm if the number 145 for reactions involved in lipid metabolism is correct.

5) Page 5, line 155: 62 in 174 is 35.6% and not 42.7%. Please confirm.

6) Page 5, line 156: 9 in 174 is 5.2% and not 31%. Please confirm.

7) Page 5, line 161: “n=43, 37.4%”, this percentage is correct if considering 115 in total, but not 230. In the bottom figure of 2B are described 230 Jdl. Please confirm.

8) Page 5, line 166: “n=22, 32.8%”, it is not clear how was calculated this percentage.

9) Page 8, line 217: “The growth is followed by the stationary phase”. Please confirm the meaning of this phrase and rephrase.

10) Page 10, line 276-278: The meaning of this sentence in the context of the tryptophan pathways as excellent targets for the discovery of new therapeutic agents in bacteria is not clear.   

Minor Comments:

1) Page 2, line 74: Please add "J2315" after "Burkholderia cenocepacia" to be clear that the GEM is from the same strain of the RNA-seq data.

2) Figure 2B: The name of several pathways is not completely written. Please correct.

3) References: Ref. 7 and 10 have doi duplicated.

4) References: Ref. 13, 15 and 22, pleases correct these references.

Reviewer 3 Report

This work describes the use of a GEM of Burkholderia cenocepacia for exploring two different purposes. Critical genes and Differential expression analysis of RNA-Seq data.

The work is somewhat straightforward, though fundamental errors were detected in the study design and at least a critical missing reference for FBA. Overall the manuscript should provide more references to back up their claims.

Global appreciation

The authors group exchange reactions together with transport reactions wrongly. The former should not be included in the study as these are pseudo-reactions used to unbalance the model and perform simulations. These are not related to the organism's metabolism, but rather to the simulation environmental conditions. This is a fundamental error as the exchange reactions are identified as "critical reactions" though they are not de facto reactions.

Regarding the results, the authors conclude that reactions in the lipids metabolism are the most represented group. However, in the original model proteins represent nearly 50% of the biomass composition, RNA 20.5%, and lipids represent almost 17% of the cell's biomass. Moreover, the original model was validated with three media, including two rich media, including amino acids and nucleobases, though not lipids. Hence, the authors' results are to be expected, as the model has to synthesize lipids, whereas other metabolites may be uptake. This analysis should have been discussed by the authors when assessing their results. Likewise, the original model development strategy also explains why the aminoacids uptake is mandatory for the model to have growth, which should be discussed.

Finally, the transcriptomics data integration analysis is impaired by the lack of information on the environmental conditions.

Major comments:

Line 072: The simulation environmental conditions should be included in the methods sections.

Line 077: The model refinement by manual curation procedure should be detailed. If the authors produced a different model, it should be provided as supplemental material.

Line 184: Which and how were the fluxes adjusted?

Line 197: Pathways within each metabolism should be detailed. A new column including the metabolism should be added in table S2.

Line 269: The analysis of critical genes should include a thorough analysis of the drug targets' inhibitors. Otherwise, the study is incomplete, as the link between both analysis is not clear.

A few sentences need to be rewritten and clarified:

Lines 106 to 111.

Lines 160 to 163.

Lines 170 to 172.

Lines 217 to 219.

Typo

Line 86: preformed

Minor comments:

The link in lines 75 and 76 does not seem to work.

Round 2

Reviewer 2 Report

The authors aswered to all the coments made to the manuscript and altered accordingly the text. Therefore, the current manuscript version is aceptabale for publication in Metabolites.

Author Response

We appreciate the reviewer's positive feedback.

Reviewer 3 Report

The authors addressed and clarified all my major concerns.

However, I still think the authors group exchange reactions together with transport reactions incorrectly. I agree with the authors that the “exchange reaction is identified as critical, it means that the uptake or secretion of the corresponding metabolite is vital for the growth of the bacteria”. However, classifying these as reactions instead of environmental conditions is erroneous.

These pseudo-reactions rely on transport reactions. Hence, the uptake or secretion of metabolites can be inferred from the critical uptake reactions, as shown in the case of the oxygen in which both the transport reaction and the exchange pseudo-reaction were identified as essential. Nevertheless, since there are cases in which metabolites are processed outside the cell, the author might want to consider performing a separate analysis, analyzing the minimal media composition through the exchange pseudo-reactions.
